# Evaluation of the Quantitative and Structural Antimicrobial Activity of Thymol, Terpinen-4-ol, Citral, and E-2-Dodecenal, Antibiotic Molecules Derived from Essential Oils

**DOI:** 10.3390/antibiotics14121202

**Published:** 2025-12-01

**Authors:** Paco Noriega, Kimberly Jaramillo, Ivana Villegas, Karla Vizuete, Ema Rivera, Alexis Debut

**Affiliations:** 1Group of Research and Development in Science Applied to Biological Resourses, Universidad Politécnica Salesiana, Avenida 12 de Octubre N2422 y Wilson, Quito 170109, Ecuador; kjaramillol2@est.ups.edu.ec (K.J.); ivillegas@ups.edu.ec (I.V.); eriverap@ups.edu.ec (E.R.); 2Department of Life Sciences and Agriculture, Universidad de las Fuerzas Armadas—ESPE, Sangolquí 171103, Ecuador; ksvizuete@espe.edu.ec (K.V.); apdebut@espe.edu.ec (A.D.)

**Keywords:** E-2-dodecenal, essential oil metabolites, antimicrobial activity, electron microscopy

## Abstract

**Background/Objectives**: This research was conducted to evaluate the antimicrobial activity of four molecules present in essential oils (thymol, terpinen-4-ol, citral, and E-2-dodecenal), complementing the study with the observation of structural damage caused by the contact of these compounds with microorganisms. **Methods**: The micro dilution in plates method was used to determine the minimum inhibitory concentration, using different concentrations of metabolites in contact with the microorganisms. Optical Microscopy was used to observe structural damage in yeasts, while Scanning Electron Microscopy (SEM) was used for bacteria. **Results**: In determining the minimum inhibitory concentration, very good activity was observed for all microorganisms at concentrations below 500 µg/mL or 0.05% *w*/*w*. In microscopic tests, we can observe three consequences of contact with the molecule to a greater or lesser extent. First, there is a clear decrease in the concentration of microorganisms. Second, we observe damage to the cell membrane. Finally, there are structural changes within the cytoplasm. **Conclusions**: This study demonstrated that the four metabolites possess good antimicrobial activity, in some of the tests they were even very close to the control antibiotics’ activity. Structural observations show that the activity can be explained by several factors. Many essential oils contain some of the molecules used, so their presence in nature could be a marker of antimicrobial activity.

## 1. Introduction

Throughout history, microbial, viral, or fungal infections have attacked humans [1]; many of them have remained latent and are common or seasonal processes, while in more complicated cases they have ended up becoming pandemics that have caused millions of deaths [1,2,3,4,5,6,7]. The most recent pandemic of COVID-19 highlighted the vulnerability we have as a species to microorganisms, causing immediate and long-term effects on physical and mental health, and on the economy of societies [8,9,10,11,12]. In addition, the pandemic caused a mortality ranging from approximately 5.5 million to 18 million deaths, depending on official data or estimates [13,14,15].

Since the discovery of penicillin by Alexander Fleming almost a century ago [16,17], antibiotics have become the preferred therapy in all health systems around the world in the treatment of infections of diverse nature [18,19]. This widespread use of antibiotics has produced a phenomenon of the global resistance of microorganisms [20,21,22,23], significantly reducing the effectiveness of antimicrobial effects and leading pharmaceutical companies to need new effective molecules every day [24,25,26].

Before the discovery of antibiotics, many infections were treated with natural products showing antimicrobial action that contain secondary metabolites such as polyphenols [27,28,29], alkaloids [30,31], terpenes [32,33,34], anthroquinones [35,36], and coumarins [37], among others.

Essential oils are a group of secondary metabolites present in the volatile fraction of various plant organs, whose main components are terpenes (especially monoterpenes and sesquiterpenes) and aromatic compounds (benzene derivatives) [38]. Among the molecules that come from essential oils with a high antimicrobial activity we have thymol [39,40,41], terpinen-4-ol [42,43], citral [44,45,46,47], and E-2-dodecenal [48,49,50].

Thymol is found in several essential oils, the best-known being thyme (*Thymus vul-garis*) and oregano (*Origanum vulgare*) [51,52,53], and the antibacterial activity of thymol has been demonstrated in several studies [54], as has its antifungal activity [55,56]. Tea tree (*Melaleuca alternifolia*) essential oil contains considerable amounts of terpinen-4-ol [57,58], a monoterpene which is also abundant in species such as *Juniperus communis* [59] and rosemary (*Salvia rosmarinus*) [60]. Terpinen-4-ol is also a molecule with significant antibacterial activity [61] and antifungal potential [62]. Citral is a mixture of two geometric isomers, neral and geranial [63]. Two plants with a high content of these molecules are *Cymbopogon citratus* [45] and *Aloysia trypylla* [64]. Essential oils with high concentrations of citral have been shown to be effective on resistant bacteria such as Helicobacter pylori [65]. Finally, there is the E-2-dodecenal molecule, which is abundant in the Amazonian plant *Eryngium foetidum* [66], used in digestive disorders; with regard to this molecule, antimicrobial studies are few, but there is a study conducted by Kubo et al. in 2003 [67], which highlights its activity against *Saccharomyces cerevisiae*. In 2004, the same author verified its good antimicrobial activity against *Salmonella* sp. [68].

In this study, the antimicrobial activity of each of the mentioned molecules was evaluated against an important number of bacteria and yeasts to corroborate their antibiotic potential; additionally, the damage caused to the microorganism by contact with the molecule was determined by Electron Microscopy.

## 2. Results

### 2.1. Antimicrobial Activity

#### 2.1.1. Determination of the Minimum Inhibitory Concentration in Gram-Positive Bacteria

The results were diverse, depending on the molecule evaluated and the microorganism. The values determined did not exceed 500 µg/mL, meaning that the resulting activity was very good. Table 1 shows the results obtained.

#### 2.1.2. Determination of the Minimum Inhibitory Concentration in Gram-Negative Bacteria

Gram-negative bacteria have a lower sensitivity; however, in the strains evaluated (*Proteus vulgaris*, *Escherichia coli*, and *Klebsiella oxytoca*), we can still find very good values of activity below 500 µg/mL. Table 2 describes the antibacterial behavior against Gram-negative bacteria.

#### 2.1.3. Determination of the Minimum Inhibitory Concentration in Yeasts

In the case of the two yeasts, both are observed to be very sensitive. Their sensitivity is less than 500 µg/mL, which means that their activity is very good. Table 3 describes the antibacterial behavior of yeasts. (Appendix A detail the procedure for determining the MIC).

### 2.2. Structural Damage to Bacteria upon Contact with the Secondary Metabolites

Figure 1 and Figure 2 show what happens when each metabolite comes into contact with bacteria. For this test, a Gram-positive and a Gram-negative bacterium were chosen, taking into account those that had a very satisfactory minimum inhibitory concentration and were easy to grow. The bacteria observed in the study are *Streptococcus mutans* (ATCC 25175) and *Klebsiella oxytoca* (ATCC 8724). In both bacteria subjected to the secondary metabolite, it is possible to observe both damage to the cell membrane and the destruction of the interior of the microorganism, which would confirm the activity of the four molecules in inhibiting the growth and proliferation of pathogens.

### 2.3. Structural Damage to Yeast upon Contact with Secondary Metabolites

Figure 3 shows significant damage to the structure of the yeast strain tested, *Candida tropicalis* ATCC 13803. There is evidence of a decrease in yeast concentration, compaction of cellular material, and disappearance of the microorganism’s membrane. The effects are observable in the four molecules used in this study. (Appendix A show individual photographs of the bacteria in contact with the bioactive molecules).

## 3. Discussion

Three of the four metabolites (thymol, citral, and terpinen-4-ol) have been studied for their antimicrobial activity.

In the scientific literature, there are several studies on the antimicrobial activity of thymol. One study shows the following MIC values: *Staphylococcus aureus* from 300 to 600 µg/mL; *Escherichia coli* from 310 to 5000 µg/mL; and *Lactobacillus* sp. from 5000 to 10,000 µg/mL [44]. A more extensive study presented by Falcone et al. in 2005 yielded the following MIC values: *Bacillus cereus* 327 µg/mL; *Bacillus subtilis* 422 µg/mL; *Bacillus licheniformis* 422 µg/mL; *Lactobacillus curvatus* 743 µg/mL; *Lactobacillus plantarum* 941 µg/mL; *Candida lusitaniae* 307 µg/mL; *Pichia subpelliculosa* 422 µg/mL; and *Saccharomyces cerevisiae* 337 µg/mL [69]. Guarda et al., 2011, obtained the following values: *Staphylococcus aureus* 250 μg/mL; *Listeria innocua* 250 μg/mL; *Escherichia coli* 250 μg/mL; *Saccharomyces cerevisiae* 125 μg/mL, and *Aspergillus niger* 250 μg/mL [70]. Finally, Sim et al., 2019, determined the following MIC values: *Pseudomonas aeruginosa* 400–800 µg/mL; *Streptococcus* sp. 200 µg/mL; *Proteus mirabilis* 200 µg/mL; and *Staphylococcus pseudintermedius* 100–200 µg/mL [71].

With regard to the antimicrobial activity of essential oils with high concentrations of thymol, *Thymus pulegoides* (24% thymol) has the following results: *Streptococcus pyogenes* 1250 µg/mL; *Staphylococcus aureus* 1250 µg/mL; *Escherichia coli* 5000 µg/mL; *Salmonella typhimurium* 10,000 µg/mL; *Pseudomonas aeruginosa* 20,000 µg/mL; *Candida albicans* 1250 µg/mL; and *Candida parapsilosis* 1250 µg/mL [72]. For Oregano vulgaris (19% thymol), the following values are obtained: *Bordetella bronchiseptica* 60–125 µg/mL; *Staphylococcus aureus* ATCC 125 µg/mL; *Escherichia coli* 500 µg/mL; and *Candida albicans* 1000 µg/mL [73]. The MIC values for thymol are quite similar to those obtained in this study, while the antimicrobial activity of essential oils is lower.

The terpinen-4-ol molecule has been shown to be active against *Enterococcus faecalis* with an MIC of 2500 µg/mL and *Fusobacterium nucleatum* 500 µg/mL [74]; another study supports its effectiveness against *Streptococcus agalactiae* with an MIC value of 98.94 µg/mL [62]. A study conducted by Noumi et al. determined activity against *Chromobacterium violaceum* and *Pseudomonas aeruginosa* with MIC values close to 50 µg/mL [75]. Finally, antifungal activity was confirmed in *Coccidioides posadasii* 350–5720 µg/mL and *Histoplasma capsulatum* 10–5720 µg/mL [76].

Antimicrobial activity studies conducted on Melaleuca alternifolia essential oil and terpinen-4-ol show the following MIC values: *S. aureus* tea tree 5000 µg/mL; terpinen-4-ol 2500 µg/mL; *E. coli* tea tree 2500–5000 µg/mL; and terpinen 4-ol 1200–2500 µg/mL, with greater activity observed in the pure molecule [77].

A study on essential oil containing 31% terpinen-4-ol verified its activity on various microorganisms, such as *Escherichia* coli 8000 µg/mL, *Staphylococcus aureus* 2000 µg/mL, *Pseudomonas aeruginosa* 12,000 µg/mL, *Penicillium italicum* Wehmer 12,000 µg/mL, and *Penicillium digitatum* Sacc. 24,000 µg/mL [78].

Several studies highlight the antimicrobial capacity of citral. Dai et al. evaluated its activity against *Staphylococcus aureus* with an MIC of 2500 µg/mL [46]. Adukwu et al. determined MIC values against *Acinetobacter baumannii* of 1400 µg/mL and *Staphylococcus aureus* of 280 µg/mL [79]. In a study conducted with *Candida albicans* yeast, they found an MIC of 64 µg/mL [76]. Extensive research carried out on various microorganisms shows the following MIC results: *Aspergillus niger* 180 µg/mL; *Trichoderma viride* 265 µg/mL; *Yersinia enterocolitica* 200 µg/mL; *Cronobacter sakazakii* 540 µg/mL; *Staphylococcus aureus* 695 µg/mL; *Bacillus cereus* 400 µg/mL; and *Enterobacter cloacae* 1000 µg/mL [80].

The *Cymbopogon citratus* essential oil with high citral content demonstrated high antimicrobial effectiveness with the following MIC results: *Escherichia coli* 128 µg/mL; *Klebsiella pneumoniae* 124 µg/mL; *Staphylococcus aureus* 64 µg/mL; and *Enterococcus faecalis* 64 µg/mL [81]. An essential oil of *Aloysia citrodora* with a citral content of 34% presented the following MIC values: *Enterobacter cloacae* 1600 µg/mL; *Escherichia coli* 1600 µg/mL; *Pseudomonas aeruginosa* 25,000 µg/mL; *Bacillus cereus* 800 µg/mL; *Listeria monocytogenes* 3100 µg/mL; and *Staphylococcus aureus* 3100 µg/mL [82].

For the molecule E-2-dodecenal, there is little information available on its antimicrobial activity. The only studies found are on the essential oil of *Eryngium foetidum*, a plant native to the American tropics, which contains a high concentration of the molecule. A study conducted on its essential oil shows MIC values in *Staphylococcus aureus* of 1000–2000 µg/mL, with the oil containing 44% E-2-dodecenal [83]. Another trial has shown activity against *Listeria monocytogenes*, *Helicobacter pylori*, *Escherichia coli*, *Pseudomonas aeruginosa*, *Staphylococcus aureus*, and *Streptococcus pneumoniae* [83].

Structural damage assessment tests performed by microscopy show very severe damage to the yeast *Candida tropicalis*, with a decrease and destruction of cells, a significant reduction in their size, and an exit of the cytoplasm. In the Gram-positive bacteria *Streptococcus mutans*, the assay shows a decrease in bacterial concentration and destruction of the membrane and cytoplasm, which is particularly noticeable upon contact with E-2-dodecenal. Finally, in the Gram-negative bacteria *Klebsiella oxytoca*, changes in the cytoplasm and the onset of damage to the membrane are observed, which are more evident upon contact with E-2-dodecenal.

## 4. Materials and Methods

### 4.1. Reagents and Microorganisms

The following chemical standards were used: 98.5% thymol Merck, code T0501; terpinen-4-ol, primary reference standard Merck, code 03900590; citral, analytical standard Supelco, code 43318; geranial and neral racemic mixture; and 95% E-2-dodecenal Sigma Aldrich, code 30658. In Figure 4, we can see the structures of the four metabolites.

The microorganisms were used were *Listeria grayi* ATCC 25401, *Streptococcus mutans* ATCC 25175, *Staphylococcus saprophyticus* ATCC 15305, *Staphylococcus aureus* ATCC 29213, *Proteus vulgaris* ATCC 6380, *Escherichia coli* ATCC 25922, *Klebsiella oxytoca* ATCC 8724, *Candida tropicalis* ATCC 13803, and *Candida albicans* ATCC 10231.

### 4.2. Antimicrobial Activity by Microdilution Plate Assay

Antimicrobial activity was performed following the protocol described by Noriega et al., 2023, for essential oils [84], with modifications in the concentrations of secondary metabolites, since this study does not work with mixtures of molecules as in an essential oil.

The process began with the preparation of the microorganism inoculum at concentrations of 10^8^ CFU/mL for bacteria (7.5 × 10^7^ CFU/mL in the wells after deposition) and 10^6^ CFU/mL for yeasts (5.05 × 10^7^ CFU/mL in the wells after deposition). The medium used for activation was Mueller–Hinton agar, the temperature for activating bacteria was 37 °C, and for yeast it was 25 °C, without modification of pH.

The inoculum was placed in the microplate wells in a volume of 150 µL, along with 20 µL of a metabolite solution in descending concentrations and 30 µL of 1% *w*/*w* 2,3,5-Triphenyl-tetrazolium chloride reagent (TTC). Secondary metabolites and the positive control were dissolved in DMSO at the following concentrations: 10%, 5%, 2.50%, 1.25%, 0.63%, 0.31%, 0.16%, 0.08, and 0.04%. The incubation time was 24 h, after which the absorbances were read in an Epoch Biotek microplate reader at a wavelength of 405 nm. Chloramphenicol was used as a positive control for bacterial activity and clotrimazole for yeast activity. The negative control was executed bacteria and TTC dye without the secondary metabolite.

The choice of microorganisms was primarily based on the availability of bacteria and yeasts in the laboratory, verification of adequate growth of the microorganisms, and having a homogeneous representation of Gram-positive bacteria, Gram-negative bacteria, and yeasts.

### 4.3. Observation of Damage in the Bacteria Using Scanning Electron Microscopy (SEM)

Several studies have been conducted to observe damage to bacterial structures using Scanning Electron Microscopy (SEM), some of them on metabolites derived from essential oils such as terpinen-4-ol and citral [46,55].

The test sample was prepared using Gram-positive *Streptococcus mutans* bacteria and Gram-negative *Klebsiella oxytoca* bacteria at a concentration of 10^8^ CFU/mL. Bacteria without the addition of any molecules were used as a negative control, while the tests were performed with the addition of chemical molecules at a concentration of 150 μg/mL; 5 μL of the bacterial culture from each treatment was added to a copper grid (formvar/carbon, 300 mesh) and then stained with 1% phosphotungstic acid (PTA, pH = 7) for one second. The samples were observed in a field emission Scanning Electron Microscope (FEG-SEM, TESCAN MIRA 3, Brno, Czech Republic) using a transmission detector.

### 4.4. Observation of Damage in Yeast Using Optical Microscopy

Yeasts can be observed using Optical Microscopy due to their large size [71], without the need for Electron Microscopy, simply by using a staining reagent to create a contrast that allows for proper observation.

The control inoculum of *Candida tropicalis* was prepared at a concentration of 10^6^ CFU/mL, then each of the molecules under study was added separately until a concentration of 150 µg/mL was reached, methylene blue was added to each as a staining reagent for 5 min, and the samples were observed under a fluorescence optical microscope (NIKON ECCLIPSE NI-U, Nikon Corporation, Tokyo, Japan) using Mshot Image Analysis System software version V 1.6.6.

### 4.5. Statistic

For the evaluation of minimum inhibitory concentration (MIC) values, relative standard deviations and statistical significance. All computations were made using the statistical software STATISTICA 6.0.

## 5. Conclusions

The antimicrobial results in the four secondary metabolites show significant activity, for most tests between 500 µg/mL (0.05%) and 50 µg/mL (0.005%), concentrations that could be used in pharmaceutical and cosmetic products, as well as food preservatives.

The antimicrobial activity of molecules such as thymol and terpinen-4-ol had already been verified, while fewer studies have been conducted on citral and E-2-dodecenal.

Although the results are not as effective as those of the control antibiotics, they are still produced at low concentrations, suggesting that they could be incorporated into systemic formulations such as capsules or syrups and, above all, into topical antimicrobial and antifungal formulations. Another interesting application could be to prevent the growth of bacteria and yeasts in processed foods and cosmetic products, through their use as preservatives, to extend the shelf life of products without causing harm to human health, replacing synthetic antimicrobials such as parabens.

The structural damage caused by the four molecules confirms the good activity observed in the microplate dilution test. The results of the research serve to propose an antimicrobial use for these molecules derived from essential oils and to suggest the continuous evaluation of these metabolites with new strains of microorganisms.

## Figures and Tables

**Figure 1 antibiotics-14-01202-f001:**
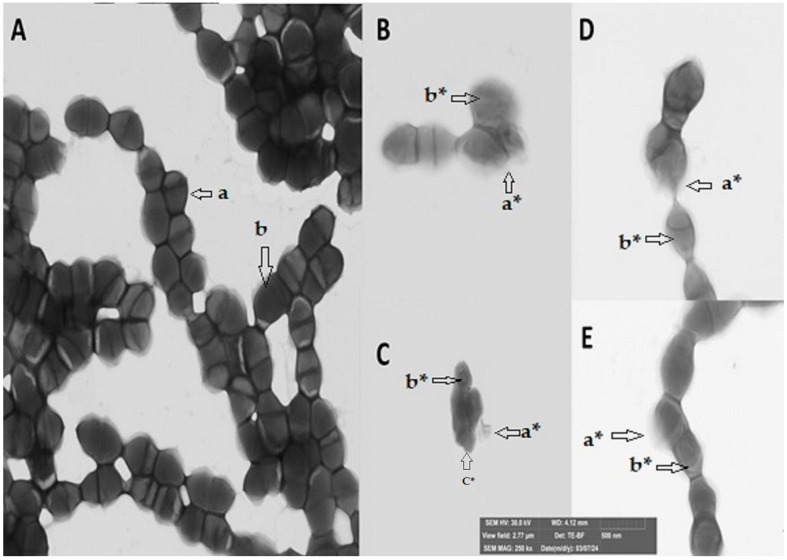
Structural damage of the Gram-positive bacterium *Streptococcus mutans* upon contact with secondary metabolites of essential oils in Scanning Electron Microscopy (SEM): (**A**) bacteria without contact with any metabolite, (**B**) in contact with citral, (**C**) in contact with E-2-dodecenal, (**D**) in contact with terpinen-4-ol, and (**E**) in contact with thymol. a undamaged cell membrane, b undamaged cytoplasm; a* damaged cell membrane, b* damaged cytoplasm, and c* decreased cell size. The arrows indicate the location of the damage caused by the metabolite.

**Figure 2 antibiotics-14-01202-f002:**
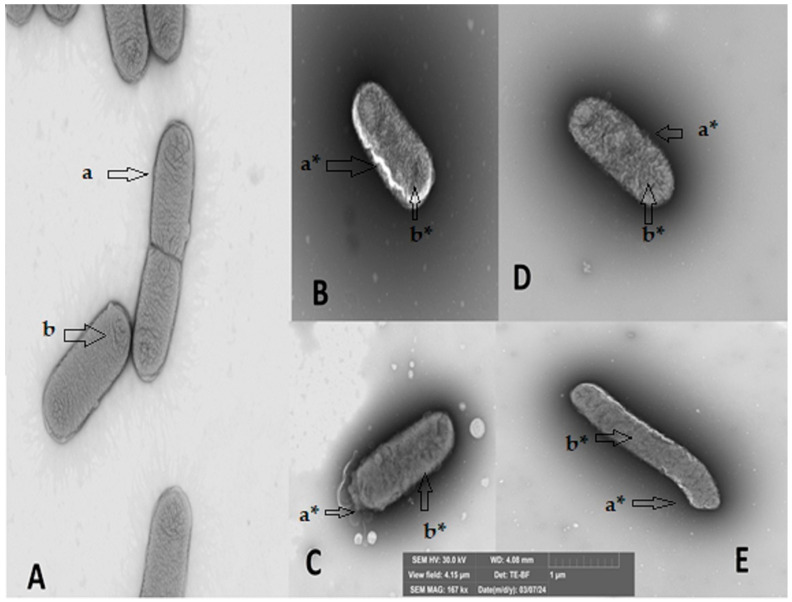
Structural damage of the Gram-negative bacterium *Klebsiella oxytoca* upon contact with secondary metabolites of essential oils in Scanning Electron Microscopy (SEM). (**A**) Bacteria without contact with any metabolite, (**B**) in contact with citral, (**C**) in contact with E-2-dodecenal, (**D**) in contact with terpinen-4-ol, and (**E**) in contact with thymol. a undamaged cell membrane, b undamaged cytoplasm; a* damaged cell membrane, and b* damaged cytoplasm. The arrows indicate the location of the damage caused by the metabolite.

**Figure 3 antibiotics-14-01202-f003:**
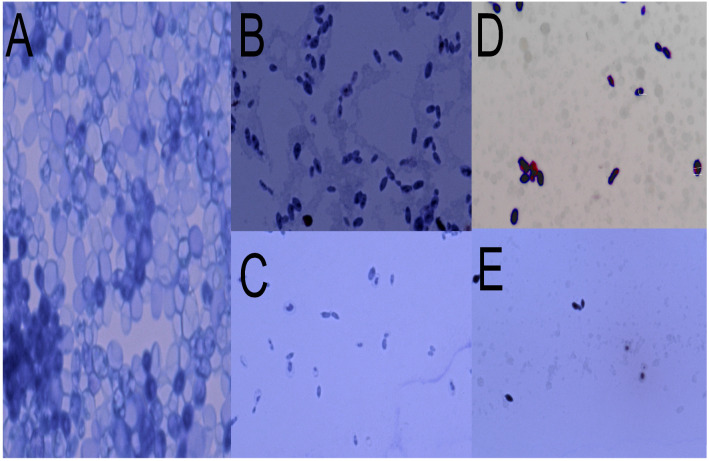
Structural damage of the yeast *Candida tropicalis* upon contact with secondary metabolites of essential oils in Optical Microscopy. (**A**) Yeast without contact with any metabolite, (**B**) in contact with citral, (**C**) in contact with E-2-dodecenal, (**D**) in contact with terpinen-4-ol, and (**E**) in contact with thymol.

**Figure 4 antibiotics-14-01202-f004:**
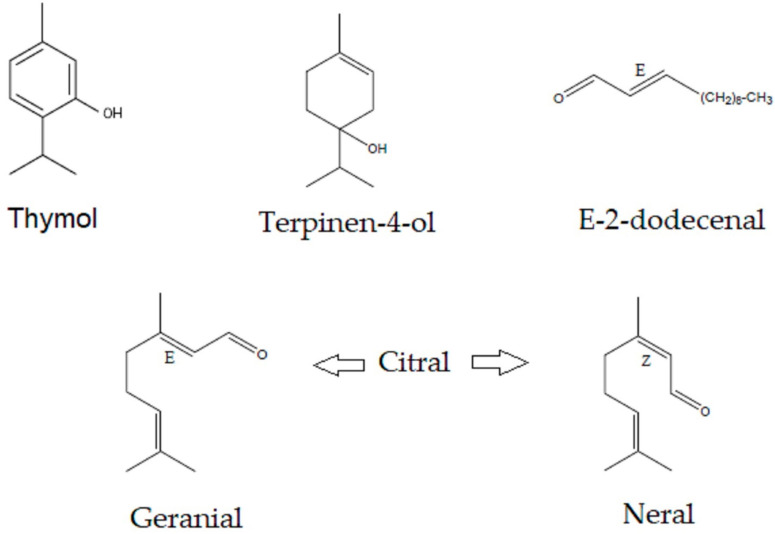
Molecules evaluated in different antimicrobial tests. (E) trans isomer, (Z) cis isomer, and O oxygenated group.

**Table 1 antibiotics-14-01202-t001:** Minimum inhibitory concentration values in µg/mL in Gram-positive bacteria. Mean ± SD (n = 3), *p* < 0.05.

Minimum Inhibitory Concentration in µg/mL					
Gram-Positive Bacteria	E-2-Dodecenal	Terpinen-4-ol	Thymol	Citral	Positive Control (Chloramphenicol)
*Listeria grayi*	332 ± 7	86.5 ± 8	91.4 ± 4	59 ± 2	18 ± 3 × 10^−3^
*Streptococcus mutans*	20 ± 0.5	142.9 ± 8	61.0 ± 3	163 ± 7	18 ± 3 × 10^−3^
*Staphylococcus saprophyticus*	15 ± 0.7	536.0 ± 25	129.0 ± 8	214 ± 11	9 ± 5 × 10^−3^
*Staphylococcus aureus*	342 ± 17	17.1 ± 1	209.0 ± 10	26.6 ± 1	18 ± 3 × 10^−3^

**Table 2 antibiotics-14-01202-t002:** Minimum inhibitory concentration values in µg/mL in Gram-negative bacteria. Mean ± SD (n = 3), *p* < 0.05.

Minimum Inhibitory Concentration in µg/mL					
Gram-Negative Bacteria	E-2-Dodecenal	Terpinen-4-ol	Thymol	Citral	Positive Control (Chloramphenicol)
*Proteus vulgaris*	257 ± 13	76.4 ± 3	97.4 ± 4	69 ± 3	40± 3 × 10^−3^
*Escherichia coli*	133 ± 6	152 ± 7	152 ± 5	76 ± 2	40 ± 3 × 10^−3^
*Klebsiella oxytoca*	192.5 ± 9	156 ± 8	89 ± 3	483 ± 18	80 ± 8 × 10^−3^

**Table 3 antibiotics-14-01202-t003:** Minimum inhibitory concentration values in µg/mL in yeasts. Mean ± SD (n = 3), *p* < 0.05.

Minimum Inhibitory Concentration in µg/mL					
Yeasts	E-2-Dodecenal	Terpinen-4-ol	Thymol	Citral	Positive Control (Clotrimazole)
*Candida tropicalis*	76 ± 3	179 ± 8	239 ± 11	125 ± 6	59 ± 4 × 10^−3^
*Candida albicans*	171 ± 9	190 ± 8	190 ± 7	209 ± 10	25 ± 6 × 10^−3^

## Data Availability

The original contributions presented in this study are included in the article/Appendix A. Further inquiries can be directed to the corresponding author.

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
