# Peer review of "Evaluation of the Quantitative and Structural Antimicrobial Activity of Thymol, Terpinen-4-ol, Citral, and E-2-Dodecenal, Antibiotic Molecules Derived from Essential Oils"

_antibiotics, 2025, doi:10.3390/antibiotics14121202_

Round 1
Reviewer 1 Report
Comments and Suggestions for Authors
This manuscript evaluated the quantitative and structural antibacterial activities of four antibacterial molecules derived from essential oils (thymol, terpinen-4-ol, citral, and E-2-dodecenal). The study determined the minimum inhibitory concentrations of these molecules against various bacteria and yeasts by the microdilution method, and observed their damage to the microbial structure using optical microscopy and scanning electron microscopy. The results showed that these molecules exhibited significant antibacterial activity at low concentrations and could cause severe damage to the cell membranes and cytoplasm of microorganisms, demonstrating their potential as natural antibacterial agents.
- In Figures 1 and 2, have exhibited structural damage of the Gram-positive bacterium Streptococcus mutans and Gram-negative bacterium Klebsiella oxytocaupon. However, the images are not clear enough. No specific characteristics of the injury were provided (such as "cell membrane rupture", "cytoplasm leakage", "cell morphology shrinkage"). Therefore, appropriate annotations and guidance can be provided on the images to help readers understand.
- The article conducted research on the antibacterial activity of four types of bacteria, namely Listeria grayi, Streptococcus mutans, Staphylococcus saprophyticus, and Staphylococcus aureus. Please provide a further explanation as to why these four types of bacteria were chosen as the research subjects.
- The conclusion section did mention the antibacterial activities of the four molecules, but the discussion on their antibacterial applications was not sufficient. It is suggested that an analysis of the application potential of these molecules in fields such as food preservation and medicine should be added.
- Ensure consistent use of terms throughout the text. For example, "E-2-docenal" and "E-2-dodecenal" should represent the same substance in different forms, and the terms must be standardized.
- In line 63 of the text, it is mentioned that "The antibacterial research on E-2-dodecenal is scarce. This study is the first to assess its antibacterial activity", but the breakthroughs in the research on E-2-dodecenal (such as the first determination of its MIC values for various strains, and the first observation of its mechanism of damage to cell structure) were not emphasized in this article. Moreover, due to the lack of data, the antibacterial advantages of this compound could not be demonstrated, and the innovation points were obscured.
- Lack of microbial culture conditions: In Section 4.2 of the article, the culture conditions for the strains were not specified (such as the type of culture medium, whether the bacteria were cultured in LB medium or other media; the culture temperature, 37℃ or other temperatures, etc.). The unclear culture conditions made it impossible to repeat the experiments and the content lacked operability.
- In the steps of Section 4.4, methyl blue was added as the staining reagent to each sample, but the specific staining time was not specified.
- The references in Articles 44 and 62 are exactly the same, with multiple repetitions and similar content being cited.
Author Response
| Observation | Correction |
|
In Figures 1 and 2, have exhibited structural damage of the Gram-positive bacterium Streptococcus mutans and Gram-negative bacterium Klebsiella oxytocaupon. However, the images are not clear enough. No specific characteristics of the injury were provided (such as "cell membrane rupture", "cytoplasm leakage", "cell morphology shrinkage"). Therefore, appropriate annotations and guidance can be provided on the images to help readers understand.
|
Signs are added to Figures 1 and 2 to differentiate between a) cell membrane, b) cytoplasm, and c) reduction in size. The signs are explained in the text at the bottom of the figures. |
| The article conducted research on the antibacterial activity of four types of bacteria, namely Listeria grayi, Streptococcus mutans, Staphylococcus saprophyticus, and Staphylococcus aureus. Please provide a further explanation as to why these four types of bacteria were chosen as the research subjects. |
The choice of microorganisms was primarily based on the availability of bacteria and yeasts in the laboratory, verification of adequate growth of the microorganisms, and having a homogeneous representation of Gram-positive bacteria, Gram-negative bacteria, and yeasts. At the end of section 4.2, a paragraph has been added that explains the choice of microorganisms in general terms. |
| The conclusion section did mention the antibacterial activities of the four molecules, but the discussion on their antibacterial applications was not sufficient. It is suggested that an analysis of the application potential of these molecules in fields such as food preservation and medicine should be added. |
Two paragraphs are added to comply with the reviewer's request, which are: Although the results are not as effective as those of the control antibiotics, they are still produced at low concentrations, suggesting that they could be incorporated into systemic formulations such as capsules or syrups and, above all, into topical antimicrobial and antifungal formulations. Another interesting application could be to prevent the growth of bacteria and yeasts in processed foods and cosmetic products as preservatives to extend the shelf life of products without causing harm to human health, replacing synthetic antimicrobials such as parabens. |
|
Ensure consistent use of terms throughout the text. For example, "E-2-docenal" and "E-2-dodecenal" should represent the same substance in different forms, and the terms must be standardized.
|
The correct name of the molecule is E-2-dodecenal. The entire manuscript has been corrected. |
|
In line 63 of the text, it is mentioned that "The antibacterial research on E-2-dodecenal is scarce. This study is the first to assess its antibacterial activity", but the breakthroughs in the research on E-2-dodecenal (such as the first determination of its MIC values for various strains, and the first observation of its mechanism of damage to cell structure) were not emphasized in this article. Moreover, due to the lack of data, the antibacterial advantages of this compound could not be demonstrated, and the innovation points were obscured. |
Considering the observations also made by reviewer 3, we were able to locate two studies on E-2-dodecenal conducted by Kubo et al. in 2003 and 2004, which have been analyzed in the introduction, thus considering two previous investigations of activity in two strains of bacteria and yeasts. This provides a comparative starting point rather than starting from scratch, as suggested by the reviewer |
|
Lack of microbial culture conditions: In Section 4.2 of the article, the culture conditions for the strains were not specified (such as the type of culture medium, whether the bacteria were cultured in LB medium or other media; the culture temperature, 37℃ or other temperatures, etc.). The unclear culture conditions made it impossible to repeat the experiments and the content lacked operability. |
Depending on the observations, the text of section 4.2 was added and modified. In the end, the section was as follows. 4.2 Antimicrobial activity by microdilution plate assay Antimicrobial activity was performed following the protocol described by Noriega et al. 2023 for essential oils [78], with modifications in the concentrations of secondary metabo-lites, since this study does not work with mixtures of molecules as in an essential oil. The process began with the preparation of the microorganism inoculum at concentrations of 10⁸ CFU/mL for bacteria (7.5 x 107 CFU/mL in the wells after deposition) and 10⁶ CFU/mL for yeasts (5.0 5 x 107 CFU/mL in the wells after deposition), The medium used for activation was Mueller-Hinton agar, the temperatures for activating bacteria were 37°C and for yeast 25°C, without modification of pH. The inoculum was placed in the microplate wells in a volume of 150 µL, along with 20 µL of a metabolite solution in descending concentrations and 30 µL of 1% w/w 2,3,5-Triphenyl-tetrazolium chloride reagent (TTC). Secondary metabolites and positive control were dissolved in DMSO at the following concentrations: 10%, 5%, 2.50%, 1.25%, 0.63%, 0.31%, 0.16%, 0.08, and 0.04%. The incubation time was 24 hours, after which the absorbances were read in an Epoch Biotek microplate reader at a wavelength of 405 nm. Chloramphenicol was used as a positive control for bacterial activity and clotrimazole for yeast activity. The negative control was executed bacteria and TTC dye without the secondary metabolite. The choice of microorganisms was primarily based on the availability of bacteria and yeasts in the laboratory, verification of adequate growth of the microorganisms, and having a homogeneous representation of Gram-positive bacteria, Gram-negative bacteria, and yeasts.
|
|
In the steps of Section 4.4, methyl blue was added as the staining reagent to each sample, but the specific staining time was not specified.
|
The staining time information, which was 5 minutes, is included.
|
| The references in Articles 44 and 62 are exactly the same, with multiple repetitions and similar content being cited. | The error indicated by the reviewer has been corrected. |
Reviewer 2 Report
Comments and Suggestions for Authors
Review comment for antibiotics-3940174 entitled “Evaluation of the quantitative and structural antimicrobial ac- 2
tivity of thymol, terpinen-4-ol, citral, and E-2-dodecenal, antibi- 3 otic molecules derived from essential oils” described by Noriega et al.
This manuscript shows antibiotic activity of essential oils for gram positive and netgative bacteria and yeast. This findings are important for clinical and sanitary fields. However, there are some serious scientific lacks in the manuscript.
1. Clarify inoculum units and final concentration. The authors state “10^8 CFU (bacteria), 150 μL per well, total 200 μL” but do not specify CFU/mL vs CFU per well.
If 10^8 CFU/mL in 150 μL: final = 7.5 x 10^7 CFU/mL
If 10^8 CFU/well: final = 5 x 10^8 CFU/mL
- The broth microdilution reference method specifies a final inoculum of 5 x 10^5 CFU/mL. Please explanin the rationale and validity for the bacterial inoculum, or realign with the reference method.
- For each chemical, specify solvent(s), stock concentration, final concentration in wells, and any solubilizer/emulsifier used with matched controls. These details are essential for reproducibility, especially for hydrophobic compounds.
- Please add medium (name, pH), temprature, incubation time, and pre-culture conditions used for each organisms.
- The authors mentioned a negative control (bacteria + TTC without compound), but no negative control values appear in Tables 1–3. For antimicrobial testing, paired negative and positive controls are required; please add the growth control, solvent control, and ideally no-cells blank (to check dye/solvent interference) to the tables.
- The authors state readings at 405 nm, yet no absorbance data are presented. Please provide the 24 h raw A405 value, and state the MIC call criterion.
- In Tables 1–3, SD values of 0 for chloramphenicol/cloterimazole are implausible across independent experiments. Please clarify n, the original readings, and any rounding or censoring applied.
- Please add positive control images alongside the untreated control and test compounds for both SEM and optical microscopy.
- Nomenclature and typographical issues.
-Unify the coupound name e.g. E-2-dodecenal; avoid mixed E-2-decenal/E-2-dodecenal.
-Correct ATTC to ATCC throughout.
-Italicize genus-species names e.g. Staphylococcus aureus and keep ATCC/strain codes in roman.
Author Response
| Observation | Correction |
|
Clarify inoculum units and final concentration. The authors state “10^8 CFU (bacteria), 150 μL per well, total 200 μL” but do not specify CFU/mL vs CFU per well. If 10^8 CFU/mL in 150 μL: final = 7.5 x 10^7 CFU/mL If 10^8 CFU/well: final = 5 x 10^8 CFU/mL The broth microdilution reference method specifies a final inoculum of 5 x 10^5 CFU/mL. Please explanin the rationale and validity for the bacterial inoculum, or realign with the reference method.
|
The actual bacterial concentration values are added in parentheses, as suggested by the reviewer. |
| For each chemical, specify solvent(s), stock concentration, final concentration in wells, and any solubilizer/emulsifier used with matched controls. These details are essential for reproducibility, especially for hydrophobic compounds |
All concentrations used are added by increasing the following paragraph. “The secondary metabolites and positive control were dissolved in DMSO at the following concentrations: 10%, 5%, 2.50%, 1.25%, 0.63%, 0.31%, 0.16%, 0.08, and 0.04.” |
|
Please add medium (name, pH), temprature, incubation time, and pre-culture conditions used for each organisms. |
Depending on the observations, the text of section 4.2 was added and modified. In the end, the section was as follows. 4.2 Antimicrobial activity by microdilution plate assay Antimicrobial activity was performed following the protocol described by Noriega et al. 2023 for essential oils [78], with modifications in the concentrations of secondary metabo-lites, since this study does not work with mixtures of molecules as in an essential oil. The process began with the preparation of the microorganism inoculum at concentrations of 10⁸ CFU/mL for bacteria (7.5 x 107 CFU/mL in the wells after deposition) and 10⁶ CFU/mL for yeasts (5.0 5 x 107 CFU/mL in the wells after deposition), The medium used for activation was Mueller-Hinton agar, the temperatures for activating bacteria were 37°C and for yeast 25°C, without modification of pH. The inoculum was placed in the microplate wells in a volume of 150 µL, along with 20 µL of a metabolite solution in descending concentrations and 30 µL of 1% w/w 2,3,5-Triphenyl-tetrazolium chloride reagent (TTC). Secondary metabolites and positive control were dissolved in DMSO at the following concentrations: 10%, 5%, 2.50%, 1.25%, 0.63%, 0.31%, 0.16%, 0.08, and 0.04%. The incubation time was 24 hours, after which the absorbances were read in an Epoch Biotek microplate reader at a wavelength of 405 nm. Chloramphenicol was used as a positive control for bacterial activity and clotrimazole for yeast activity. The negative control was executed bacteria and TTC dye without the secondary metabolite. The choice of microorganisms was primarily based on the availability of bacteria and yeasts in the laboratory, verification of adequate growth of the microorganisms, and having a homogeneous representation of Gram-positive bacteria, Gram-negative bacteria, and yeasts.
|
|
In Tables 1–3, SD values of 0 for chloramphenicol/cloterimazole are implausible across independent experiments. Please clarify n, the original readings, and any rounding or censoring applied.
|
In fact, the SD data for the positive control were rounded up because they were very small. At the reviewer's request, they are included as exponentials so as not to have too many decimal places. |
|
The authors mentioned a negative control (bacteria + TTC without compound), but no negative control values appear in Tables 1–3. For antimicrobial testing, paired negative and positive controls are required; please add the growth control, solvent control, and ideally no-cells blank (to check dye/solvent interference) to the tables. |
In the specific case of this research, the negative control was used qualitatively (verifying the intense reddish coloration of TTC when there is no activity). The method on which this test, published in the journal Moléculas (Noriega et al. 2023), and other previous tests are based is founded on this premise. |
|
The authors state readings at 405 nm, yet no absorbance data are presented. Please provide the 24 h raw A405 value, and state the MIC call criterion. |
The microplate method involves absorbance readings at nine concentration positions ranging in our case from 10% (10,000 ug/mL) to 0.04% (40 ug/mL) for the solutions. To this must be added the three repetitions, five molecules, and the calibration curve, followed by the calculation of the MIC. It was not included in the manuscript due to the size of the information, as there are about 20 pages of data and calculations. We suggest that the reviewer prepare a specific supplementary document for each type of microorganism, three in total (Gram +, Gram-, and yeast), so that it can be read by researchers and better describe the technique. |
| Please add positive control images alongside the untreated control and test compounds for both SEM and optical microscopy. | In Figures 1, 2, and 3, photographs (A) are those of microorganisms controlled by SEM and optical microscopy. Original photographs are included in Supplementary Document 2 for better reader comprehension, informing the reviewer of this. |
|
Nomenclature and typographical issues. |
All nomenclature errors identified by the reviewer are corrected. The correct name of the molecule is E-2-dodecenal. |
Reviewer 3 Report
Comments and Suggestions for Authors
This article summarizes authors’ evaluation of the antimicrobial activities of various terpenoids. Authors also present their findings on the action of these natural molecules on the bacterial and yeast cell membrane.
The antimicrobial activity of the terpenoids and their action on the cell membrane have been widely documented. Many of these literature findings were presented by the authors in the manuscript. The authors claim that this is the first study conducted on the antimicrobial activity of E-2-dodecenal (Lines 246-247). However, this is not valid as the following article reported the anti-salmonella activity of E-2-dodecenal:
- Kubo, K. Fujita, K. Nihei, A. Kubo, Anti‐Salmonellaactivity of (2E)‐alkenals, Journal of Applied Microbiology, Volume 96, Issue 4, 1 April 2004, Pages 693–699, https://doi.org/10.1111/j.1365-2672.2003.02175.x
Antifungal activity was also reported:
- Kubo I, Fujita KI, Kubo A, Nihei KI, Lunde CS. Modes of antifungal action of (2 E)-alkenals against Saccharomyces cerevisiae. Journal of agricultural and food chemistry. 2003 Jul 2;51(14):3951-7.
Moreover, following articles on the antimicrobial activity of essential oils where E-2-dodecenal is the main compound diminishes the novelty of the work:
- Silalahi M. Essential oils and uses of Eryngium foetidum L. GSC Biological and Pharmaceutical Sciences. 2021;15(3):289-94.
- Kumar S, Ahmad R, Saeed S, Azeem M, Mozūraitis R, Borg-Karlson AK, Zhu G. Chemical composition of fresh leaves headspace aroma and essential oils of four Coriander cultivars. Frontiers in Plant Science. 2022 Feb 17;13:820644.
**Authors use ‘E-2-docenal’ throughout manuscript, however compound is named as ‘E-2-dodecenal’ in Figure 4, which I believe is the correct one.
In its current form and given the findings, the lack of previous studies on the compounds, and the lack of innovation, the article is not suitable for publication in the journal Antibiotics.
Author Response
| Observation | Correction |
|
The antimicrobial activity of the terpenoids and their action on the cell membrane have been widely documented. Many of these literature findings were presented by the authors in the manuscript. The authors claim that this is the first study conducted on the antimicrobial activity of E-2-dodecenal (Lines 246-247). However, this is not valid as the following article reported the anti-salmonella activity of E-2-dodecenal:
Antifungal activity was also reported:
Moreover, following articles on the antimicrobial activity of essential oils where E-2-dodecenal is the main compound diminishes the novelty of the work: Silalahi M. Essential oils and uses of Eryngium foetidum L. GSC Biological and Pharmaceutical Sciences. 2021;15(3):289-94. Kumar S, Ahmad R, Saeed S, Azeem M, Mozūraitis R, Borg-Karlson AK, Zhu G. Chemical composition of fresh leaves headspace aroma and essential oils of four Coriander cultivars. Frontiers in Plant Science. 2022 Feb 17;13:820644 |
With regard to the molecule E-2-dodecenal, the information suggested by the reviewer has been incorporated with the two studies proposed by Kubo et al. Despite this, it can still be said that antimicrobial research with this molecule is scarce. The antimicrobial study on Eryngium foetidum oil, conducted by Silalahi in 2021, has already been considered in the manuscript, and Kumar's 2022 study was not mentioned as it only evaluates the chemical composition and does not conduct antimicrobial studies. An important consideration of our research group is that the E-2-dodecenal molecule is still novel enough to warrant further antibiotic research. |
|
Authors use ‘E-2-docenal’ throughout manuscript, however compound is named as ‘E-2-dodecenal’ in Figure 4, which I believe is the correct one. |
The correct name of the molecule is E-2-dodecenal. The entire manuscript has been corrected. |
Round 2
Reviewer 2 Report
Comments and Suggestions for Authors
The authors generally responded appropriately to my points. I believe this manuscript deserves acceptance. Congratulations to the authors.
Author Response
Dear Reviewer, thank you very much for your insightful comments, which have improved our manuscript.
Reviewer 3 Report
Comments and Suggestions for Authors
Although important publications regarding the work done previously on the antimicrobial activity of E-2-dodecenal, raised by the Reviewer previously, added by the authors, they continue to claim that this is the first work on the antimicrobial effect of this compound (Lines 256-258).
In order to claim a decrease in microbial count in a culture, microscopy images are not sufficient, CFU count experiments must be performed. In order to claim cell membrane disruption and leakage of the cellular material; leakage of cellular material (e.g. 280 nm absorbing material test), conductivity, PI uptake experiments must be conducted. Given data are not sufficient.
Line 121, Candida tropicalis is not bacteria.
Author Response
Observation 1
The description in lines 256-258 is amended, removing the statement that this is a first study for E-2-dodecenal.
Observation 2
The reviewer's observation is timely. Unfortunately, in our laboratory we do not have the suggested tests implemented, such as: cellular material (e.g. 280 nm absorbing material test), conductivity, PI uptake experiments must be conducted and others.
Observation 3
The error on line 121 has been corrected, recognizing C. tropicalis as yeast.
Round 3
Reviewer 3 Report
Comments and Suggestions for Authors
As explained in the previous report, the data presented does not warrant the conclusions derived in the manuscript. Authors state that they do not have the necessary infrastructure to perform the suggested tests. However, CFU counting does not require any equipment, but human eye to count colonies; 280 nm absorbing material testing needs a spectrophotometer a basic instrument of all microbiology settings.
Author Response
Dear Reviewer, We appreciate your observations. Regarding the requested assay, we have been evaluating your suggestions. However, the strains used have lost viability, so we believe that performing the assay would yield data inconsistent with the initial research. Similarly, we believe that the spectrophotometric determination of the microdilution method already indicates a decrease in bacterial load, albeit in a different way. Therefore, we request that the manuscript be considered valid, as two other reviewers have already done.
Thank you for your attention.
Best regards
